# Morphology of Composite Fe@Au Submicron Particles, Produced with Ultrasonic Spray Pyrolysis and Potential for Synthesis of Fe@Au Core–Shell Particles

**DOI:** 10.3390/ma12203326

**Published:** 2019-10-12

**Authors:** Peter Majerič, Darja Feizpour, Bernd Friedrich, Žiga Jelen, Ivan Anžel, Rebeka Rudolf

**Affiliations:** 1Faculty of Mechanical Engineering, University of Maribor, Smetanova ulica 17, 2000 Maribor, Slovenia; z.jelen@um.si (Ž.J.); ivan.anzel@um.si (I.A.); rebeka.rudolf@um.si (R.R.); 2Zlatarna Celje d.o.o., Kersnikova 19, 3000 Celje, Slovenia; 3Institute of Metals and Technology, Lepi pot 11, 1000 Ljubljana, Slovenia; darja.feizpour@imt.si; 4IME Process Metallurgy and Metal Recycling, RWTH Aachen University, Intzestrasse 3, 52065 Aachen, Germany; bfriedrich@ime-aachen.de; 5Plastika Skaza d.o.o., Selo 22, 3320 Velenje, Slovenia

**Keywords:** Ultrasonic Spray Pyrolysis, core–shell nanostructures, Fe@Au, iron oxide particles, Au nanoparticles, nanoparticle morphology

## Abstract

Iron core–gold shell (Fe@Au) nanoparticles are prominent for their magnetic and optical properties, which are especially beneficial for biomedical uses. Some experiments were carried out to produce Fe@Au particles with a one-step synthesis method, Ultrasonic Spray Pyrolysis (USP), which is able to produce the particles in a continuous process. The Fe@Au particles were produced with USP from a precursor solution with dissolved Iron (III) chloride and Gold (III) chloride, with Fe/Au concentration ratios ranging from 0.1 to 4. The resulting products are larger Fe oxide particles (mostly maghemite Fe_2_O_3_), with mean sizes of about 260–390 nm, decorated with Au nanoparticles (AuNPs) with mean sizes of around 24–67 nm. The Fe oxide core particles are mostly spherical in all of the experiments, while the AuNPs become increasingly irregular and more heavily agglomerated with lower Fe/Au concentration ratios in the precursor solution. The resulting particle morphology from these experiments is caused by surface chemistry and particle to solvent interactions during particle formation inside the USP system.

## 1. Introduction

Iron oxide nanoparticles are one of the most commonly used magnetic nanoparticles [1]. In order to give them other functional properties for applications, their surfaces are modified with different materials, such as polymers, organic monolayers, oxides and metals [1,2]. A common modification is a gold shell on the oxide particles, since gold enhances light scattering and absorbance of the resulting particles due to its surface plasmon resonance, has possibilities for the conjugation of other functional groups, is biocompatible [1,3,4] and has unique catalytic properties in nanoparticle form [5,6]. Composite iron core–gold shell nanoparticles are researched widely for their broad uses in medical treatment [3,7], magnetic resonance imaging [8], cancer treatment [9] and drug delivery systems, as well as for catalysis, sensors, and so on [1,10,11,12], due to the coupling of magnetic and optical properties of iron and gold, when in nanoparticle form. The combination of these properties makes these particles of interest for use in energy, solar cells and fuel cells [6].

Given the potential uses of Iron core–gold shell (Fe@Au) nanoparticles, several synthesis routes and production methods have been used to make them, from chemical routes to laser ablation and others [3,9,10,13,14,15,16,17]. Our investigation aims to determine the possibility of producing Fe@Au core–shell nanoparticles with Ultrasonic Spray Pyrolysis (USP). USP is a well-known powder production method, which is also capable of producing nanoparticles. Compared to other nanoparticle production methods, USP has the advantage of continuous production on a large scale, making it a commercially competitive process. Another advantage of such a one-step synthesis is core–shell nanoparticle production without the use of organic linkers, which can alter the final particle properties and interactions [16].

USP uses the dissolved salt of a material in a precursor solution, aerosolises it into droplets via ultrasound, and then produces particles from each droplet at high temperatures with a reaction gas. We have used this method successfully to produce gold nanoparticles (AuNPs), the synthesis of which was conceptually based on the Turkevich method [18,19]. As such, the precursor solution was prepared with chloroauric acid (HAuCl_4_), which was aerosolised into droplets and reduced with hydrogen gas in the USP system, producing Au nanoparticles. In an effort to examine the possibilities of producing core–shell nanoparticles with USP, we have produced Fe@Au nano- and submicron particles experimentally, using combinations of several Fe and Au precursors [20]. 

The precursors used for Fe were iron (II) acetate, iron (III) chloride and iron (III) nitrate, while the precursors used for Au were gold (III) acetate, gold (III) chloride and gold (III) nitrate. This showed that the best results with USP were produced with a precursor combination of iron chloride and gold chloride. However, the resulting particles did not have a core–shell structure, but were rather submicron Fe oxide particles decorated with Au nanoparticles. As this was an initial investigation for the synthesis of Fe@Au core–shell particles, it was proposed that the ratio between Fe and Au precursors was not enough to cover the Fe oxide particle with a uniform coating of Au [20]. Another set of experiments was carried out in order to confirm this suggestion. In addition, these experiments also clarified the exact way in which these particles are formed from the single aerosol droplet inside the USP process.

## 2. Materials and Methods 

The precursor solutions for producing the Fe@Au particles were prepared with different concentrations of iron (III) chloride hexahydrate (trace metals basis ≥98%, Molekula) and gold (III) chloride tetrahydrate (trace metals basis ≥99.9%, Acros Organics). The concentrations are shown in Table 1. The concentrations were selected based on our previous experience with USP and the goal to achieve nanosized particles. For this reason, the concentrations of individual precursors were kept at a maximum of 1 g/L. The precursor solutions were prepared by dissolving the salts in 1 L of deionised water. Thermogravimetric analysis (TGA) was performed on a PerkinElmer TGA 4000 System (PerkinElmer, Waltham, MA, USA). TGA of the iron (III) chloride hexahydrate was performed to determine the reaction temperatures in the USP device, while TGA of the gold (III) chloride was performed in our previous investigations and checked from the literature [21,22].

The USP device consists of an ultrasonic aerosol generator, a tube furnace and an electrostatic filter for particle collection. The device is shown in Figure 1. The ultrasonic generator Gapusol 9001 (RBI, France) has three ultrasonic transducers with a frequency of 2.5 MHz. The tube furnace has three heating zones, with a length of 0.4 m, 1 m and 0.4 m (pre-heating, reaction and cooling, respectively), with a temperature range of 0–1100 °C. The quartz tube inside the tube furnace is 1.8 m long, with a diameter of 42 mm. The whole system was under a low vacuum of around 980–990 mbar. The precursor solutions were put in the ultrasonic generator chamber, where they were aerosolised. Nitrogen and hydrogen gases were passed through the ultrasonic generator. Nitrogen carried the precursor aerosol through the tube furnace, where reactions occurred with the hydrogen. Nitrogen gas flow was set to 4 L/min, while the hydrogen gas flow was set to 2 L/min. The synthesised particles were then deposited on the grid in the electrostatic filter. The electrostatic filter was also heated up to 150 °C to prevent re-condensation of the water vapours before the gas carried them out of the system.

### Characterisation

Scanning and Transmission Electron Microscopy (SEM and TEM) was used for the micrographs and micro chemical analysis. The SEM used was Sirion 400NC (FEI, Hillsboro, OR, USA) with an Energy-Dispersive X-ray spectroscope INCA 350 (Oxford Instruments, Abingdon, UK). Powder samples of the produced particles were collected and a volume of 0.25 mL of powder was transferred onto standard 12.5 mm carbon film holders for SEM observations.

JEOL 2100 (JEOL, Tokyo, Japan) and JEOL JEM-2200FS HR (JEOL, Japan) operating at 200 kV were used for the TEM. The collected powder samples with a volume of about 0.25 mL were dispersed in 1 mL ethanol. A drop of this suspension was put on a copper TEM grid with an amorphous carbon film. The grids were then dried before they were used for TEM investigations.

The particle sizes were measured with the microscope software, along with the Image J software [23]. The morphology (sizes, shapes and Au coverage on Fe oxide) was evaluated from the SEM and TEM images and EDX results. The particles were collected dry in an electrostatic filter. As such, DLS and zeta potential were not measured.

The powder X-ray diffraction (XRD) measurement was carried out using an X’Pert PRO Powder X-ray diffractometer 3040 (PANalytical, Almelo, Netherlands), operating at 45 kV, 40 mA, with Cu Kα1 radiation (λ = 1.54060 Å) in the 2θ range from 5° to 90° with the 0.008° step per 99,695 s. A quantity of 2 mL of the powder sample was prepared on a zero-background Si holder. The X’Pert High Score Plus program was used to identify the crystal structure and the phases present in the sample. 

## 3. Results

TGA of the obtained iron (III) chloride hexahydrate showed a 76% total weight loss for decomposition above 500 °C (Figure 2). This was the starting point to determine the USP reaction temperature. A temperature of 600 °C was selected to ensure that total decomposition would occur during the USP process. For the gold (III) chloride tetrahydrate, TGA was already observed from other USP experiments with this compound and from the literature [21,22]. Its decomposition temperature is around 275 °C, while the selected USP reaction temperature of 600 °C is enough for the particle synthesis reactions to be carried out.

Figure 3 shows the particles produced by the four different precursor concentrations. Additional images are presented in the supporting information for a better overview of the particles produced (Appendix A and TEM Bright Field images in Appendix A). In the backscattered images, the white particles are Au, while the grey spheres are Fe oxide. This is additionally confirmed by energy-dispersive X-ray spectroscopy (EDX) analysis, and an example is shown in Figure 4. 

The crystal structure and the phases present in the produced particles from the experiment Fe/Au 2 were determined using XRD, which confirmed the presence of several iron oxides and gold, as shown in Figure 5. 

XRD confirmed the presence of wustite Fe_0.942_O (PDF-Nr. 01-073-2144), maghemite Fe_2_O_3_ (PDF-Nr. 00-025-1402) and Au (PDF-Nr. 00-004-0784). The main diffraction peaks in the spectrum are Fe_2_O_3_. Also, some magnetite Fe_3_O_4_ (PDF-Nr. 01-075-0033) may be present in the sample, but in a very small amount which could not be detected clearly by XRD and differentiated from Fe_2_O_3_. Additional main peaks of Au are present in the sample as well. Additionally, some peaks in the spectrum of Fe/Au 2 of an unknown cubic phase (Fm-3m) are present.

A distinct particle morphology and AuNP arrangement difference is seen in the images of the prepared particles. When using a low concentration of iron in the precursor solution, the resulting particles have a greater number of larger AuNPs, which are more heavily agglomerated. As the ratio increases in favour of Fe, the AuNPs retain greater numbers, but become smaller and more uniform in shape.

Experiment Fe/Au 0.1 shows Fe oxide particles with larger meshes of agglomerated AuNPs present in the samples. The measured Fe oxide particle mean size is about 258 nm, as shown in the Appendix A. The AuNPs’ sizes were about 45 nm. It is important to note that Fe oxide particle size measurement is difficult, due to the particles being covered in AuNPs, while the AuNP size measurements are to be considered an estimation, as it is not possible to determine the exact edges of the particles in the agglomerates. For this reason, mostly solitary AuNPs and particles with visible edges were measured for comparison with the other experiments.

The sample Fe/Au 0.25 shows a lesser agglomeration of the AuNPs, but a larger AuNP size than in the Fe/Au 0.1. The measured Fe oxide particle mean size was 390 nm, while the measured AuNP mean size was 67 nm. In fact, the AuNPs were not larger than in the experiment Fe/Au 0.1, there was just a much lower number of smaller AuNPs present. The size distribution in the Appendix A shows a clearer comparison regarding particle sizes.

In experiment Fe/Au 2, the Fe oxide mean particle sizes were measured to be 336 nm. The AuNP mean size was measured as 24 nm. As the Fe/Au precursor concentration ratio increases, the AuNPs become smaller and less agglomerated. Some agglomeration is still present, while the AuNPs begin to arrange themselves across the Fe oxide particle surfaces.

Increasing the Fe/Au precursor concentration ratio to 4 yields Fe oxide particles with a mean size of 374 nm. The surfaces of Fe oxide particles are decorated with more uniformly dispersed AuNPs, with a measured mean nanoparticle size of 26 nm.

A comparison of the different particle morphologies obtained from the experiments is shown in Figure 6.

## 4. Discussion

From the initial experiments reported previously [20], intended to examine the feasibility of producing Fe@Au core–shell nanoparticles with USP, we have observed fine AuNPs decorating much larger Fe oxide particle surfaces. Such structures were formed inside the USP device from the aerosol droplets with dissolved Fe and Au. From the observations in this report, we have concluded that a higher Au content as compared to Fe content in the droplets would result in a better, more uniformly shaped coating on top of the Fe oxide particles [20]. As such, we have determined the Fe/Au precursor concentration ratio in this report to be the main factor when modifying the Fe@Au particle size and morphology with this method. When changing this ratio, the overall precursor concentration should be kept low, about 1–2 g/L, for the USP to produce nanoparticles, as is evident from experience and the literature for this synthesis method [20,21,22]. There are several observations that can be made from the performed experiments and analyses. The most important observation is that merely increasing Au content as compared to Fe content in the droplets does not produce more intrinsically uniform Au coatings.

The mean sizes of Fe oxide particles are not very different across the experiments. In the lowest Fe/Au ratio of 0.1, the mean size was about 258 nm, while higher ratios produced Fe oxide particles with mean sizes of 300–400 nm. Excluding ratio 0.1, the Fe oxide size distributions are also similar. For the production of smaller Fe oxide particles, the Fe precursor concentration would need to be reduced. This would, in turn, allow the reduction of Au concentration in the precursor, as less Au would be needed to cover the smaller Fe oxide particles.

When considering the aerosol droplet with dissolved Fe and Au inside the USP, it seems that the AuNPs are formed near the droplet surface, while the Fe oxide particles are formed inside the droplet core, due to the different physico-chemical and rheological properties of the gold chloride and iron chloride (density, viscosity, surface tension, etc.). As the droplet evaporates and shrinks, the surface chemistry of the system causes the AuNPs near the droplet surface to be deposited on the Fe oxide particles inside the core of the droplet. Some cases of Fe@Au core–shell synthesis report growth of Au on the Fe core, or inverted-growth of Fe on an Au shell, depending on whether the solvent in which the growth took place was organic, water or otherwise [5,16,24]. The selection of solvent for the precursor preparation for USP synthesis may also be revised for a continuous Au shell production.

In experiments with an Fe/Au ratio of 2 and 4, the AuNPs had very similar sizes, while there was more agglomeration present, with a ratio of 2. Considering the higher Au precursor concentration with a ratio of 2, we can establish that there were more AuNPs present in the final product in this experiment, since the sizes were about the same in the experiment with a ratio of 4. The higher AuNP number resulted in increased agglomeration. In the experiment with a ratio of 4, the AuNPs were dispersed more uniformly across the Fe oxide particles. As we decreased the Fe/Au ratio, increasing the Au content in the aerosol droplet, the AuNPs grew more irregular in shape and were dispersed across the samples. However, in our goal to produce a more uniform Au coating on top of the Fe oxide particles, the higher Fe/Au ratio does not provide enough Au content in the droplet, which is needed to cover the entire Fe oxide particle surface.

Important elements to consider are the AuNP morphologies and arrangements of AuNPs on top of Fe oxide particles, and why they occur in our experiments. The adhesive forces between the Fe and Au particles seem to be much lower than the cohesive forces in the AuNPs. This results in Au clustering and clumping on top of Fe oxide particles. Fe and Au are known to have broad miscibility gaps in their phase diagrams, usually resulting in complete phase separation, which causes difficulties in trying to produce Fe–Au alloy nanoparticles [16]. As such, a continuous layer of Au on top of Fe oxide particles would not be possible to achieve without modifying the adhesive forces between the Fe oxide particles and Au. Researching and adding additives to the initial precursor solution with Fe and Au chlorides might be one possible way of changing the chemical interactions between the elements and the solvent.

For core–shell nanoparticle production, an intermediate layer between the Fe core and Au shell is usually made with functional groups (citrates, thiols, amines, etc.), facilitating continuous Au shell growth [3,10,12,14,16]. Following these methods in the case of USP, the Fe oxide particles would need to first be produced and later coated with Au. However, this would invalidate the advantages of using USP for production of these particles, as it would no longer be continuous and in one step.

The USP production parameters are also an important factor to consider for Fe@Au particle production. Changing the ultrasonic frequency modifies the initial aerosol droplet sizes, while changing the temperatures or using alternative gases may yield other iron oxides due to different formation kinetics. It is also possible to modify the particles after their formation, as shown in some reports of converting core–shell Fe_2_O_3_@TiO_2_ particles to magnetic Fe_3_O_4_@TiO_2_ particles by reduction under H_2_ flow [2]. These are further activites to be researched surrounding USP production of Fe@Au, along with a complete determination of the magnetic properties of the resulting particles.

However, the particles already produced in this work are suitable as catalysts for carbon monoxide (CO) oxidation [5,25], selective reduction of nitrogen-containing compounds [5], plasmonic heating [26] and magnetic hyperthermia [26,27]. A nano-sized gold catalyst, supported on iron oxides, can be highly effective for hydrogenation and oxidation reactions [28]. Catalysts promoting the oxidation of the toxic CO gas at ambient temperatures may be included in indoor air purification devices, gas masks, oil refining, the production of pharmaceuticals, for cleaning automotive exhaust gases, or electrocatalysts for energy conversion and storage [5,25]. The added magnetic properties can be used for recovery and reuse of the particles after catalytic reactions.

Fe_2_O_3_ particles decorated with AuNPs, with a similar morphology as the particles in this work, were also produced for plasmonic heating [26]. The presented Fe_2_O_3_-Au nanoparticles retained both magnetic and plasmonic properties. This was demonstrated by testing the magnetic behavior of the particles using a magnet, and by photothermal measurements taken while heating aqueous solutions using a laser. It was shown that the Fe_2_O_3_-Au nanoparticles could heat the solutions as efficiently as pure AuNPs, which require up to 20× higher Au concentrations for the same task [26]. The Fe_2_O_3_-Au particles thus reduce costs for plasmonic heating and have an added functionality of being able to be collected magnetically from the heated solution.

For hyperthermia, the AuNP decoration provides suitable cytotoxicity and easier surface functionalization for these particles, without changing the heating efficiency of the iron oxides for destroying tumour cells in medicine [27]. These particles, therefore, have an interesting potential for a bimodal application of light and magnetic hyperthermia. However, for biomedical imaging and photothermal therapy, the solubility of the produced Fe@Au particles in water would need to be increased by surface functionalization, as low solubility leads to aggregation and precipitation of the particles, making them less suitable for biological applications [1].

## 5. Conclusions

The Fe oxide submicron particles decorated with AuNPs were produced with USP, using a precursor solution with dissolved iron (III) chloride and gold (III) chloride with different Fe/Au concentration ratios ranging from 0.1 to 4. In a previous report, it was presumed that a lower Fe/Au ratio would result in a more uniformly shaped coating on top of the Fe oxide particles [20], due to the higher Au content during particle formation. However, the results showed that a high Fe/Au ratio produced more uniformly dispersed AuNPs with sizes about 20 nm on top of Fe oxide particles, while a low ratio produced larger and heavily agglomerated AuNPs. The cohesive forces between the AuNPs were much higher than the adhesive forces between Fe oxide and Au, resulting in AuNPs clumping on top of Fe oxide particles. Producing a continuous layer of Au on top of Fe oxide particles would not be possible to achieve with USP, without modifying the adhesive forces between the Fe oxide particles and Au during particle formation.

## Figures and Tables

**Figure 1 materials-12-03326-f001:**
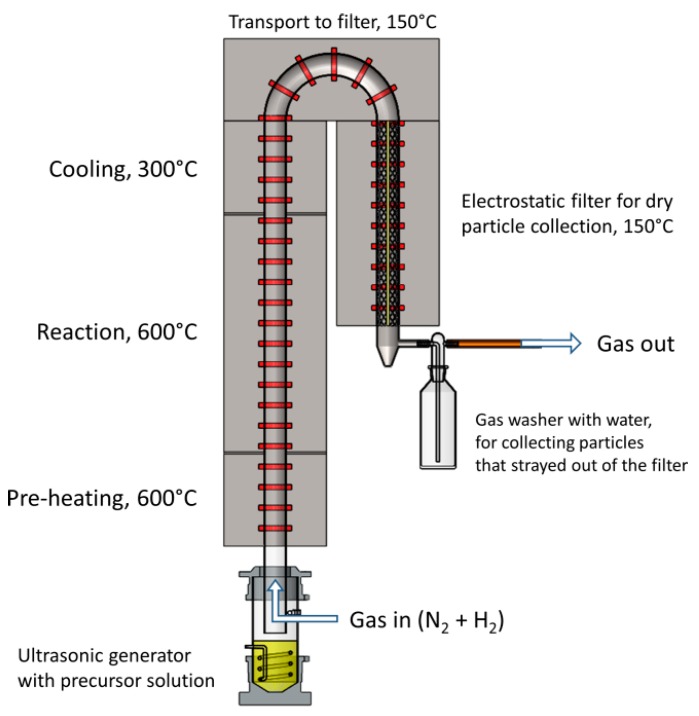
Schematic of the Ultrasonic Spray Pyrolysis (USP) device used for Fe@Au nanoparticle synthesis.

**Figure 2 materials-12-03326-f002:**
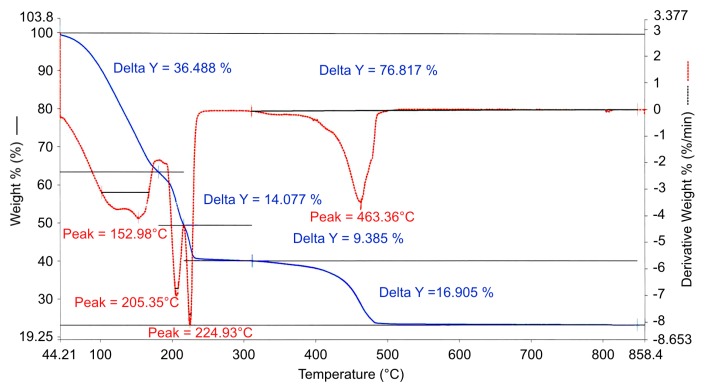
Thermogravimetric analysis of iron (III) chloride hexahydrate.

**Figure 3 materials-12-03326-f003:**
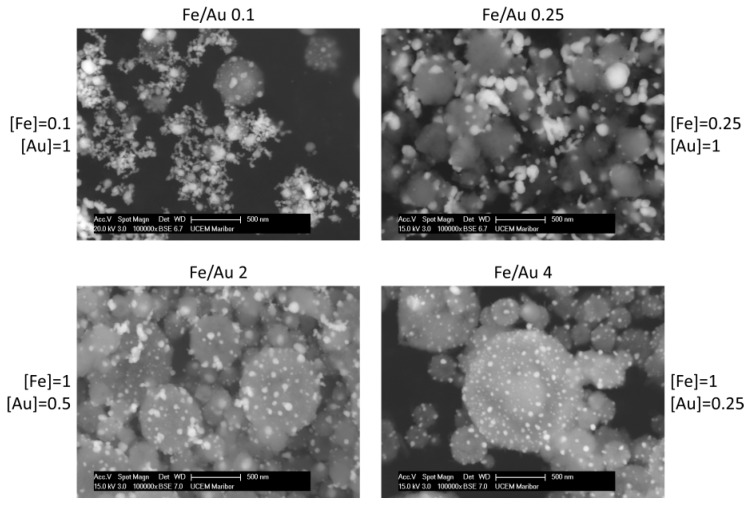
SEM images of the Fe@Au particles produced in experiments with different Fe and Au concentrations, i.e., with different Fe/Au concentration ratios. Concentrations are in g/L.

**Figure 4 materials-12-03326-f004:**
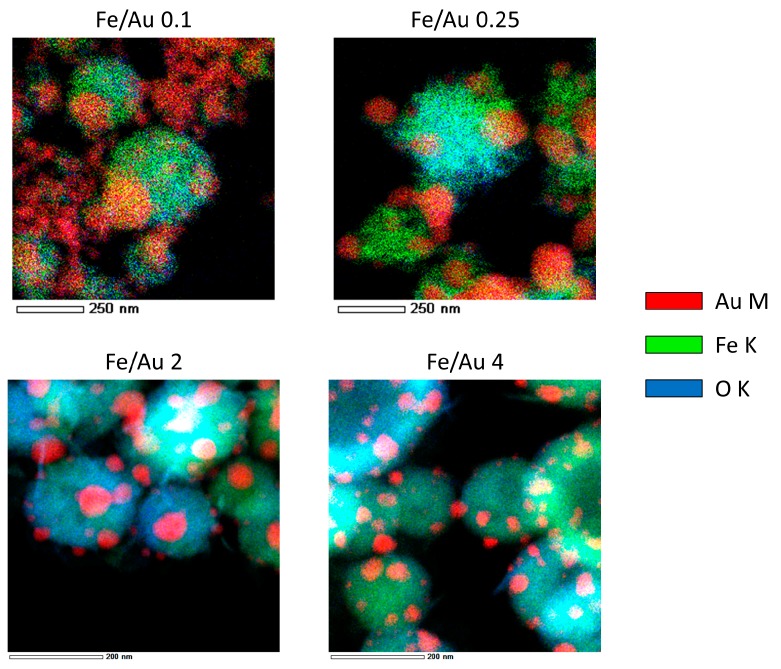
EDX mapping of the Fe@Au particles produced in experiments with different Fe/Au concentration ratios. Bright Field TEM images for the corresponding experiments are presented in Appendix A.

**Figure 5 materials-12-03326-f005:**
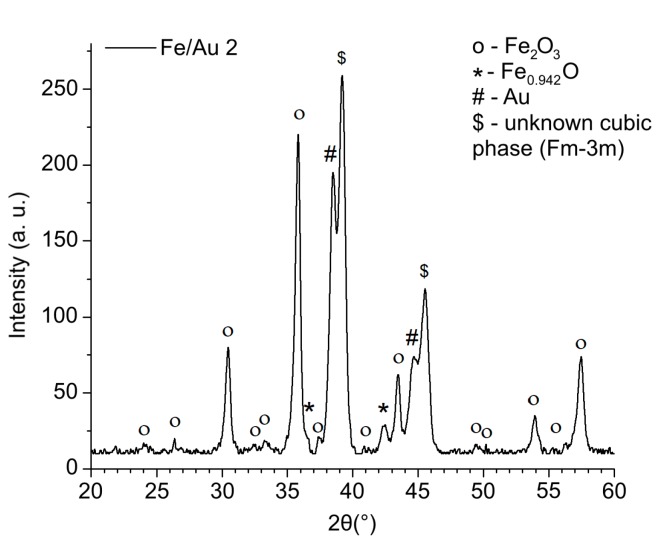
XRD pattern of the particles produced by USP from the experiment Fe/Au 2, with identified maghemite Fe_2_O_3_, wustite Fe_0.942_O, and Au crystal structures. Some unidentified peaks of an additional cubic phase (Fm-3m) are present in the spectrum.

**Figure 6 materials-12-03326-f006:**
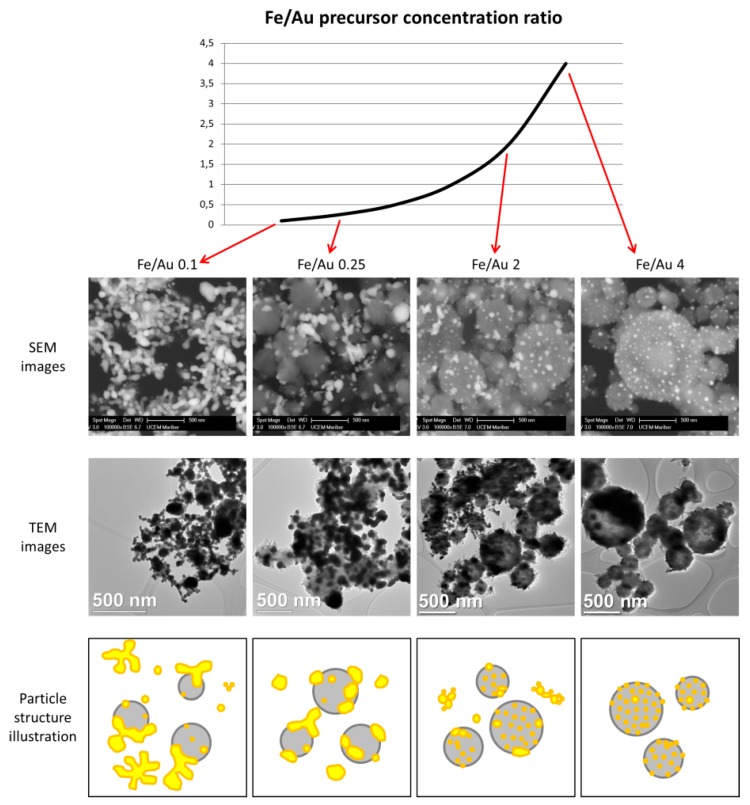
Comparison of different Fe@Au particle morphologies obtained with USP, using different Fe/Au concentration ratios in the precursor solution.

**Table 1 materials-12-03326-t001:** Table of experiments for iron core–gold shell (Fe@Au) nanoparticle synthesis performed with Ultrasonic Spray Pyrolysis (USP).

Experiment	Iron (Fe) Concentration	Gold (Au) Concentration	Fe/Au Ratio
Fe/Au 0.1	0.1 g/L	1 g/L	0.1
Fe/Au 0.25	0.25 g/L	1 g/L	0.25
Fe/Au 2	1 g/L	0.5 g/L	2
Fe/Au 4	1 g/L	0.25 g/L	4

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
