# Peer review of "Morphology of Composite Fe@Au Submicron Particles, Produced with Ultrasonic Spray Pyrolysis and Potential for Synthesis of Fe@Au Core–Shell Particles"

_materials, 2019, doi:10.3390/ma12203326_

Round 1

Reviewer 1 Report

Dear Editor,

I accurately reviewed the article

Title Morphology of composite Fe@Au submicron particles, produced with Ultrasonic Spray Pyrolysis and potential for synthesis of Fe@Au core-shell particles

submitted to Materials.

The topic is interesting and suitable for the Journal, but the authors have to solve some issues.

The Introduction is lacking on the core-shell and composite nanomaterials overview, and in particular about metal-polymer nanoparticles preparations and applications. It would be useful for the readers to have some recent references on this topic. Just as examples:

Fully Crystalline Faceted Fe–Au Core–Shell Nanoparticles 2015Nano Letters 15(8) Bioconjugation of gold-polymer core-shell nanoparticles with bovine serum amine oxidase for biomedical applications; Colloids and Surfaces B: Biointerfaces 134 (2015) 314-321 Nanoscale Architecture of Bimetallic Hybrid Fe-Au Nanostructures with and without 1,4-Phenylene Diisocyanide Pre-Functionalization 2015 RSC Advances 5(40) Y3+ embebbed in polymeric nanoparticles: morphology, dimension and stability of composite colloidal system; Colloid and Surface A 532 (2017) 125-131 Composite Titanium Dioxide Nanomaterials; Chem. Rev. 2014, 114,9853-9889 Plasmon controlled shaping of gold nanoparticles aggregates by femtosecond laser induced melting; J. Phys. Chem. Lett., 2018, 9, pp 5002–5008 Enhanced Fluorescence Emission and Magnetic Alignment Control of Biphasic Functionalized Composite Janus Particles; Volume36, Issue1, 2019, 1800311

Experimental part

More details on the quantities and concentrations used for the characterizations must be provided

Results

Have the authors thought about testing the magnetic properties?

Discussion

the authors should highlight the novelty and the application aspect of the material

Table 1 could be moved in Supporting Material

English must be reviewed a few sentences are too long

In conclusion, the article will be published only after major revisions.

best regards

Author Response

Point 1: The Introduction is lacking on the core-shell and composite nanomaterials overview, and in particular about metal-polymer nanoparticles preparations and applications. It would be useful for the readers to have some recent references on this topic. Just as examples:

Fully Crystalline Faceted Fe–Au Core–Shell Nanoparticles 2015Nano Letters 15(8) Bioconjugation of gold-polymer core-shell nanoparticles with bovine serum amine oxidase for biomedical applications; Colloids and Surfaces B: Biointerfaces 134 (2015) 314-321 Nanoscale Architecture of Bimetallic Hybrid Fe-Au Nanostructures with and without 1,4-Phenylene Diisocyanide Pre-Functionalization 2015 RSC Advances 5(40) Y3+ embebbed in polymeric nanoparticles: morphology, dimension and stability of composite colloidal system; Colloid and Surface A 532 (2017) 125-131 Composite Titanium Dioxide Nanomaterials; Chem. Rev. 2014, 114,9853-9889 Plasmon controlled shaping of gold nanoparticles aggregates by femtosecond laser induced melting; J. Phys. Chem. Lett., 2018, 9, pp 5002–5008 Enhanced Fluorescence Emission and Magnetic Alignment Control of Biphasic Functionalized Composite Janus Particles; Volume36, Issue1, 2019, 1800311

Response 1: The core-shell and composite nanomaterials overview was expanded with a focus on Iron oxide-Gold composite particles, with added references. Some references from the suggested literature was also added. Since the paper focuses on Iron oxide particles, with a surface modification of Gold nanoparticles, the metal-polymer nanoparticles preparations and applications were not included, as the authors believe that readers could go elsewhere for this information.

Experimental part

Point 2: More details on the quantities and concentrations used for the characterizations must be provided

Response 2: Added details about the quantities of the powder samples used for characterizations. The changes to the manuscript are colored in red.

Results

Point 3: Have the authors thought about testing the magnetic properties?

Response 3: The paper is more focused on the possibilities of producing Fe@Au core-shell nanoparticles, while the magnetic properties will be discussed in future works. The discussion was extended with the possibilities for further work with USP along with magnetic properties and the application of the produced nanomaterials.

Discussion

Point 4: the authors should highlight the novelty and the application aspect of the material

Response 4: Added a paragraph discussing about the application of the produced nanomaterials.

Point 5: Table 1 could be moved in Supporting Material

Response 5: The presentation of precursor solution concentrations used in the experiments and the Fe/Au ratios is an important aspect for the experimental section, since this results in the various morphologies of the produced particles.

Point 6: English must be reviewed a few sentences are too long

Response 6: Some sentences were changed. English was also proofread by a native speaker. The changes to the manuscript are colored in red.

Reviewer 2 Report

The manuscript "Morphology of composite Fe@Au submicron particles, produced with Ultrasonic Spray Pyrolysis and potential for synthesis of Fe@Au core-shell particles", written by Majerič et al., describes a preparation of FexOy@Au nanocomposites using USP process. The theme is interesting and produces materials are well characterized.

I only suggest to better characterize also the magnetic properties of this nanomaterial as they can be important in further applications.

Author Response

Point 1: I only suggest to better characterize also the magnetic properties of this nanomaterial as they can be important in further applications.

Response 1: The paper is more focused on the possibilities of producing Fe@Au core-shell nanoparticles, while the magnetic properties will be discussed in future works. The discussion was extended with the possibilities for further work with USP along with magnetic properties determination and the possibilities for application of the produced nanomaterials.

Reviewer 3 Report

Dear authors,

thank you for your well-prepared manuscript. Your presentation is very interesing, clear and scientifically sound.

However, I have some formal suggestions:

Figure 2 is hard to read, it appears to be blurry. I propose to increase font size Same for Fig. 5, the indices are hardly readable

Author Response

Point 1: Figure 2 is hard to read, it appears to be blurry. I propose to increase font size Same for Fig. 5, the indices are hardly readable

Response 1: The font sizes in Figures 2 and 5 were increased. A sharper image was given for Figure 2.

Round 2

Reviewer 1 Report

Authors have solved the problems and the article is now ready for publication

Author Response

No reply needed.